# Use of Fourier Series in X-ray Diffraction (XRD) Analysis and Fourier-Transform Infrared Spectroscopy (FTIR) for Estimation of Crystallinity in Cellulose from Different Sources

**DOI:** 10.3390/polym14235199

**Published:** 2022-11-29

**Authors:** Nicolás Montoya-Escobar, Daniel Ospina-Acero, Jorge Andrés Velásquez-Cock, Catalina Gómez-Hoyos, Angélica Serpa Guerra, Piedad Felisinda Gañan Rojo, Lina Maria Vélez Acosta, Juan Pablo Escobar, Natalia Correa-Hincapié, Omar Triana-Chávez, Robin Zuluaga Gallego, Pablo M. Stefani

**Affiliations:** 1Programa de Ingeniería en Nanotecnología, Universidad Pontificia Bolivariana, Circular 1_N_70-01, Medellín 050031, Colombia; 2Electro Science Laboratory, Department of Electrical and Computer Engineering, The Ohio State University, Columbus, OH 43210, USA; 3Facultad de Ingeniería Agroindustrial, Universidad Pontificia Bolivariana, Circular 1_N_70-01, Medellín 050031, Colombia; 4Facultad de Ingeniería Química, Universidad Pontificia Bolivariana, Circular 1_N_70-01, Medellín 050031, Colombia; 5Facultad de Ingeniería de Sistemas, Universidad Pontificia Bolivariana, Circular 1_N_70-01, Medellín 050031, Colombia; 6Grupo Calidad, Metrología y Producción, Instituto Tecnológico Metropolitano, Medellín 050013, Colombia; 7Grupo de Biología y Control de Enfermedades Infecciosas (BCEI), Sede de Investigación Universitaria, Universidad de Antioquia, Medellín 050010, Colombia; 8Instituto de Investigaciones en Ciencia y Tecnología de Materiales (INTEMA), Consejo Nacional de Investigaciones Científicas y Técnicas (CONICET), Universidad Nacional de Mar del Plata (UNMdP), Av. Colón 10850, Mar del Plata 7600, Argentina

**Keywords:** cellulose, crystallinity, crystallinity index, peak deconvolution, X-ray diffraction, infrared spectroscopy, Fourier series

## Abstract

Cellulose crystallinity can be described according to the crystal size and the crystallinity index (CI). In this research, using Fourier-transform infrared spectroscopy (FTIR) and X-ray diffraction (XRD) methods, we studied the crystallinity of three different types of cellulose: banana rachis (BR), commercial cellulose (CS), and bacterial cellulose (BC). For each type of cellulose, we analyzed three different crystallization grades. These variations were obtained using three milling conditions: 6.5 h, 10 min, and unmilled (films). We developed a code in MATLAB software to perform deconvolution of the XRD data to estimate CI and full width at half-maximum (FWHM). For deconvolution, crystalline peaks were represented with Voigt functions, and a Fourier series fitted to the amorphous profile was used as the amorphous contribution, which allowed the contribution of the amorphous profile to be more effectively modeled. Comparisons based on the FTIR spectra and XRD results showed there were no compositional differences between the amorphous samples. However, changes associated with crystallinity were observed when the milling time was 10 min. The obtained CI (%) values show agreement with values reported in the literature and confirm the effectiveness of the method used in this work in predicting the crystallization aspects of cellulose samples.

## 1. Introduction

Cellulose is a high-molecular-weight polymer composed of D-glucose units linked by β1-4 glycosidic bonds. Cellulosic hydroxyl groups are involved in intra- and intermolecular hydrogen bonds and give rise to various ordered crystalline arrangements along with disordered regions [1]. Four crystalline cellulose allomorphs have been identified from their characteristic X-ray diffraction (XRD) patterns and solid-state ^13^C nuclear magnetic resonance (^13^C NMR) spectra: cellulose I, II, III, and IV [2]. Cellulose crystallinity is described according to four characteristics: the nature or arrangement of the crystal lattice, crystal size, relative orientation of crystals, and relative amounts of crystalline components. The relative amounts of crystalline components are usually described using the crystallinity index (CI), which can be measured through techniques such as XRD [2], ^13^C NMR [3], Fourier-transform infrared spectroscopy (FTIR) [4], and Fourier-transform Raman spectroscopy [5]. The level of crystallinity is strongly related to material strength, stiffness [6], and elastic modulus [7], and is also an indicator of cellulose enzymatic hydrolysis [8]. Although the specific correlation of these factors is not fully understood, accurate determination of the CI facilitates the formation of a basis for further analyses and investigations.

There are different methods used to calculate CI, which results in different estimates [9]. Some of the most popular methods are FTIR analysis (also known as the Nelson and O’Connor method [4]), the Newman method, and ^13^C NMR C4 peak separation [10]. In addition, there are a wide arrange of methodologies involving XRD characterization, for instance, the Segal [11], XRD deconvolution [12,13], and XRD amorphous subtraction [14] methods. The Segal method is the most widely adopted due to its simplicity [15]. However, it is regarded as a quick and rough estimate [16], whereas other techniques such as XRD deconvolution often offer more accurate results, as the contribution of both the amorphous and crystalline profiles is considered [2].

The XRD deconvolution method comprises mathematical treatment to deconvolute the diffraction data into separate peaks associated with crystalline planes and amorphous cellulose according to Bragg’s law. The CI is obtained through the ratio of the sum of the areas below the curve of the crystalline peaks and the area below the entire diffraction curve [2]. Traditionally, for this method, issues with accuracy have been shown, such as in CI estimation, because common peak functions such as Lorentz and Gaussian fail to produce satisfactory fitting to the amorphous profile [12]. This ultimately translates into poor performance when determining the contribution of the amorphous profile in the whole diffractogram pattern of a cellulose sample [12].

In this study, we sought to improve the accuracy of CI estimation and to more effectively determine the contribution of the amorphous profile of cellulose through fitting it with a Fourier series using MATLAB software. Then, we developed a code in MATLAB, and the Fourier-based fitted model of the amorphous profile was used as the amorphous contribution in an XRD deconvolution of the diffraction data of semicrystalline cellulose, whereas Voigt peaks were considered for the crystalline peaks such that the deconvolution could be performed to more accurately determine crystal size and CI (%).

To achieve the above, three cellulose types were studied: (i) cellulose from banana rachis, (ii) wood-derived commercial cellulose, and (iii) bacterial cellulose. For each cellulose type, three samples were prepared with different relative amounts of crystalline components via different methods: one without milling treatment, which was in the form of a thin film; one milled for 10 min; and the final (an amorphous sample) milled for 6.5 h. XRD was used to assess the cellulose crystalline structure of all samples. An XRD pattern of semicrystalline cellulose was obtained from the film sample and the sample milled for 10 min, whereas an amorphous profile was obtained from the sample milled for 6.5 h. Fourier series were used to fit the amorphous profile, which has been found to be effective in modeling the amorphous contribution for different kinds of cellulose [12]. All samples were subjected to attenuated total reflection–Fourier-transform infrared spectroscopy (ATR–FTIR) to analyze the absorption bands related to crystallinity.

## 2. Materials and Methods

### 2.1. Materials

Cellulose samples from three different sources were used in this study: cellulose obtained from banana rachis (BR), commercial sample (CS) extracted from wood, and bacterial cellulose (BC). The cellulose from BR was obtained using the method proposed by Zuluaga et al. [17] followed by mechanical homogenization in a disc mill [18]. CS was supplied by an international producer whose production method is unknown, and BC was synthesized from the *Komagataeibacter medellinensis* strain from Colombian vinegar culture according to the methodology of Castro et al. [19,20] followed by mechanical homogenization.

### 2.2. Ball-Milling Treatment

Two groups of BR, CS, and BC samples were employed. For the first group, the samples were kept unmilled in film form. Films were formed by vacuum filtration of 0.1 wt.% cellulosic suspensions through a nylon membrane of a pore size of 0.2 μm; the resultant films were oven-dried at 40 °C for 5 days and analyzed without further processing.

The second group comprised a cellulose ball milled at room temperature. Here, two subsets were considered: in the first subset, BR, CS, and BC samples were ball-milled for 10 min to reduce crystallinity [21], whereas in the second subset, all samples were ball-milled for 6.5 h to achieve complete amorphization [12], as presented in Figure 1. BR, CS, and BC samples from the first group were labeled as Film BR, Film CS, and Film BC, respectively. Specimens milled for 10 min are denoted as P_BR_BM_10 m, P_CS_BM_10 m, and P_BC_BM_10 m (Figure 1).

Ball milling was performed on a Cryomill (Retsch, Haan, Germany) at room temperature. The specifications and settings are shown in Table 1. The sample was kept at room temperature during grinding, so the cryogenic function was not used.

### 2.3. Attenuated Total Reflection–Fourier-Transform Infrared Spectroscopy (ATR-FTIR)

The spectra of all samples were obtained using a Nicolet iS50 spectrophotometer (Thermo Scientific, Waltham, MA, USA) in the range 4000–400 cm^−1^ using a diamond ATR single-bounce crystal. The spectra were recorded at a resolution of 4 cm^−1^ with accumulation of 64 scans. ATR correction was performed using Omnic software (Thermo Fisher Scientific, Waltham, MA, USA) using a refractive index of 1.493 for cellulose [22]. The baseline was automatically corrected according to the linear baseline in agreement with the software default settings, where 15-point smoothing was used to avoid the loss of absorbances of interest for analysis. Spectra were normalized to the highest absorbance for qualitative comparison. Prior to analysis, samples were dried at 40 °C for 24 h.

### 2.4. X-ray Diffraction (XRD)

Films and milled samples from the three types of cellulose were analyzed via XRD (Figure 1). One-dimensional powder XRD was performed on an Empyrean 2012 (Malvern-PANalytical, Worcestershire, United Kingdom) modular powder diffractometer (MPD) with a PIXcel3D detector in focusing geometry mode, known as Bragg–Brentano geometry or powder mode. The MPD operates at 45 kV and 45 mA. The CuKα X-ray had a wavelength of 1.5418 Å. A goniometer omega/2θ was used. The platform setting was a reflection transmission spinner rotating at 4 rpm. A range of 10° to 40° was used for the diffraction angle 2θ with an angle step of 0.0262606°/min and a scan speed of 55 s.

For the incident ray, a 0.04 Soller, 10 mm mask, 0.5° divergence slit, and 1° antiscattering slit were used. The diffracted ray passed through a 0.04 Soller and a 0.5° antiscattering slit. A monochromator and collimator were not used. A holder of 10 mm diameter and 1 mm thickness was used for the powder samples. For cellulose films, a different sample holder was implemented.

In this study, the background subtraction was made following the method proposed by Yao et al. [12]. To reduce the XRD background originating from diffraction and incoherent scattering associated with the sample holder [23,24], a zero-background holder (ZBH) made of single-crystal silicon was used because it generates low background [12]. In addition, XRD instrumentation introduces distortion of diffraction data, making diffraction peaks asymmetrical compared to the ideal symmetrical peaks predicted by Bragg’s law [25]. We corrected the polarization effect before data analysis. The factor for this correction is known to be $\left(1+cos^{2}\left(2\theta \right)\right)/2$ [26].

### 2.5. Amorphous Profile Fitting

Based on the Fourier-series concept, a function f at x can be approximated through $f\left(x\right)=a_{0}+\overset{K}{\underset{k=1}{\sum }}[a_{k}\cos\left(k\omega_{o}x\right)+b_{k}\sin\left(k\omega_{o}x\right)]$, where $a_{0},\;\omega_{o},\;a_{k},$ and $b_{k}$ are parameters to be determined in the fitting process, and the order of the series, k, is set by the user. In this work, for the diffraction spectrum of the amorphous profiles, $a_{0},\;\omega_{o},\;a_{k},$ and $\;b_{k}$ were established via an optimization process minimizing the distance (Euclidean norm) between the true (amorphous profile) values and the fitted ones. The fitting process for the amorphous profile of each sample was conducted in MATLAB using K = 8, which allowed the highest R^2^ to be obtained. The “optimal” values of the Fourier coefficients and *w_o_* along with the value of R^2^ obtained for each type of cellulose are shown in the results section.

### 2.6. Peak Deconvolution

A peak deconvolution routine was implemented in a code developed in MATLAB to calculate CI (%) and the crystal sizes of the samples of film and those milled for 10 min for the different cellulose types. This MATLAB procedure separates the XRD data into the amorphous contribution and crystalline contribution through a curve-fitting process.

The process performs linear combination of the fitted amorphous profile from the Fourier-series fitting and a number of peaks representing the crystalline contribution (each one described by Voigt peak function):(1)$$\hat{XRD}\left(2\theta ;\gamma_{1},\ldots ,\;\gamma_{n}\right)=c_{1}P_{Fourier}\left(2\theta ;\gamma_{1}\right)+c_{2}P_{V}^{\left(1\right)}\left(2\theta ;\gamma_{2}\right)+c_{3}P_{V}^{\left(2\right)}\left(2\theta ;\gamma_{3}\right)+c_{N}P_{V}^{\left(N-1\right)}\left(2\theta ;\gamma_{N}\right)$$
where:

$\hat{XRD}$ represents the estimate of the entire XRD diffractogram;2θ represents the diffraction angle, which is an independent variable;$\gamma_{n}$ represents the set of defining parameters of the n-th peak (e.g., position in horizontal axis, width, etc.);$c_{n}$ denotes the linear combination coefficient corresponding to the n-th peak;$P_{Fourier}$ represents the fitted amorphous profile of the type of cellulose under study;$P_{V}^{\left(n\right)}$ denotes the peak function describing the n-th crystalline peak and represents the total number of peaks considered for the deconvolution (with peaks forming the crystalline contribution).

The fitting process of the entire XRD profile corresponds to an optimization procedure, implemented in MATLAB with its built-in interior-point method that seeks to find ${\left\{c_{n}\right\}}_{n=1}^{N}$ and ${\left\{\gamma_{n}\right\}}_{n=1}^{N}$ such that the distance (Euclidean norm) between the actual XRD data and $\hat{XRD}$ is minimized.

For the crystalline contribution, there are two aspects that need to be defined a priori: the type of peak function (represented by $P_{V}^{\left(n\right)}$ for the n-th crystalline peak in Equation 1) and the number of peaks to be used (N−1 in Equation (1)) [2]. Regarding the number of peaks, four crystalline peaks ($1\bar{1}0,\;110,\;200,$ and 004) have been considered in previous work [27], whereas five crystalline peaks ($1\bar{1}0,\;110,\;102,\;200,$ and 004) were considered in another study [28]. We considered five crystalline peaks in this study so that we could examine as many crystalline peaks as possible without contemplating an excessive number of peaks that would increase the complexity of the deconvolution [29].

Regarding the type of function used to describe the crystalline peaks, Gaussian [30], Lorentzian [27], and Voigt [28] functions are commonly used. We adopted Voigt functions for the five crystalline peaks as these functions presented a better fit than Gaussian and Lorentzian functions [12].

It is important to note here that in the peak deconvolution process of the XRD data, while all of the parameters involved in Equation (1) could, in principle, be freely varied, some important constraints were imposed on a few of those quantities. More specifically:

The coefficients ${\left\{c_{n}\right\}}_{n=1}^{N}$ were never allowed to be negative.For the amorphous peak ($P_{Fourier}$ in Equation (1)), although its position on the horizontal axis represents an option that can be adjusted in the optimization routine, it was restricted to between 18 and 20 2θ.For the Voigt peaks, in general, since they correspond to a convolution of a Gaussian function and a Lorentzian function, in addition to their positions on the horizontal axis, both the Gaussian and the Lorentzian widths are available for adjustment by the optimization routine. In this case, the centers of the peaks were allowed to move in a range of 0–0.4 [29,31].

Once the deconvolution procedure is completed, the MATLAB code calculates CI (%) by finding the ratio between the sum of the area below all the Voigt peaks and the sum of the area below all the peaks, including the one corresponding to the amorphous contribution [13,32]. The computation of the area below the peak profiles is carried out by finely interpolating their ordinate values and then performing numerical integration via the trapezoid method.

Figure 2 shows a schematic summary of the peak deconvolution process and the CI calculation followed by the MATLAB code. For the crystal-size calculation, the code finds the full width at half-maximum (FWHM), and its value is replaced in Equation (2) [33]:(2)$$\tau =\frac{k\lambda }{\beta \cos\theta }$$
where *τ* is the crystal size, *k* is a correction factor, and *β* and *θ* (in radians) are the FWHM and the location 2θ for the peak where the crystal size is being calculated, respectively.

## 3. Results and Discussion

Infrared spectroscopy is one of the techniques most commonly used for cellulose characterization [34], and in some cases, it has been used to study chain conformation [4] and crystal structure [35]. Figure 3a–c show the infrared spectra of the films and powders of BR, CS, and BC samples. Increasing the milling time reduced the absorbances at several wavenumbers, which can be due to a significant decrease in crystallinity [4]. These absorbances are presented and analyzed in Table 2. It should be mentioned that the shape and intensity of the absorbances change, but they still appear at the same wavenumber, indicating that the chemical structure has not changed.

Figure 3d summarizes the infrared spectra of the samples P_BR_BM_6.5 h, P_CS_BM_6.5 h, and P_BC_BM_6.5 h, which are apparently in an amorphous state. This may be said because no changes are observed when the samples are compared to each other, which is consistent with the information reported in [38]. However, the samples P_BR_BM_10 m, P_CS_BM_10 m, and P_BC_BM_10 m present changes in their absorption bands which are believed to be associated to crystallinity [4].

Samples were X-rayed to confirm the results of the different samples analyzed through ATR–FTIR and for further analysis of the crystallinity changes observed at different milling times.

The results from fitting of Fourier functions to the amorphous profile of the three cellulose samples are shown in Figure 4a–c. Fourier-series expansion was used to fit the amorphous profile instead of Gaussian [39], Lorentz [27], and Voigt peak [28] functions, because there have been issues with these attempts. For the mentioned peak functions, the shape of the diffractogram differs from that of the true profile [12]. Thus, Fourier-series expansion can be used to obtain a good representation of the amorphous cellulose XRD data, resulting in fits with coefficients of determination R^2^ > 0.99 [12].

The fits of XRD profiles show a shoulder around 15–17° 2θ with a broad peak of maximum intensity around 20.47 for BR, 20.68 for CS, and 20.46 2θ for BC. Finally, between 32.5 and 40° 2θ, a broader peak is observed for all the amorphous profiles. The absence of or strong reduction in all peaks corresponding to values of the Bragg angle characteristic to cellulose I demonstrates that these samples are amorphous [36]. It is important to note that the multiple scattering maxima in the XRD data of the three amorphous cellulose profiles are attributed to additional local order in the cellulose chains, which is enhanced by intramolecular hydrogen bonding [31]. Yao et al. [12] attributed this diffractogram shape to the atomic spacings within the glucan chain or the short-range order (SRO) of cellulose, which may result in the presence of periodic or quasi-periodic atomic features in amorphous cellulose. SRO refers to d-spacings of less than 20 Å [12].

It can be observed that the diffractogram shapes of the three amorphous cellulose profiles, attributed to the SRO of cellulose [12], tend to a common profile. These similarities observed in the three amorphous profiles were expected due to the amorphous reference obtained through cryogrinding in the powder diffraction file (PDF entry 00-060-1501) published by the International Centre for Diffraction Data (ICDD) [31].

### 3.1. Amorphous Equation

For the three samples (BR, CS, and BC), an eight-order, real-form Fourier-series model (Equation 3) was used for fitting the amorphous cellulose XRD data with background subtraction and XRD intensity correction. The XRD data are available in Appendix A.
(3)$$\begin{array}{c} f\left(x\right)=a0+a1*\cos\left(x*w\right)+b1*\sin\left(x*w\right)+a2*\cos\left(2x*w\right)+b2*\sin\left(2x*w\right)+a3\\ \;\;\;\;\;\;\;*\cos\left(3x*w\right)+b1*\sin\left(3x*w\right)+a4\\ \;\;\;\;\;\;\;*\cos\left(4x*w\right)+b4*\sin\left(4x*w\right)+a5*\cos\left(5x*w\right)+b5*\sin\left(5x*w\right)+a6\\ \;\;\;\;\;\;\;*\cos\left(6x*w\right)+b6*\sin\left(6x*w\right)+a7*\cos\left(7x*w\right)+b7*\sin\left(7x*w\right)+a8\\ \;\;\;\;\;\;\;*\cos\left(8x*w\right)+b8*\sin\left(8x*w\right) \end{array}$$

In Equation (3), *x* is 2θ in radians, and the coefficients (with 95% confidence bounds) for each sample are listed in Table 3.

### 3.2. Deconvolution with Fourier Function

XRD data with background subtraction and intensity correction were analyzed using peak deconvolution. These corrections improve the R^2^ for the different samples, satisfying the expectation that these data treatments correct the experimental deviations associated with the instrument and lead to a more symmetrical XRD diffractogram that can more realistically fit the crystalline content [12].

Shown in Figure 5, Figure 6 and Figure 7 are the XRD data and deconvolution for the films (a) and the powders milled for 10 min (b). In cellulose of land plants as BR and CS, the predominant crystal form is monoclinic Iβ [21,35]. For both specimens, the three main peaks for the Iβ monoclinic unit cell have Miller indices of ($1\bar{1}0$), (110), and (200) [40], and their centers are around 14.8, 16.6, and 23° 2θ, respectively. There are two other peaks with Miller indices (102) and (004) and centers around 20.6 and 34.6° 2θ, respectively, whose reflection is more clearly distinguished in the 10 min milled powder of each sample. The better resolution of these peaks may be due to the random orientation of the crystallites in the powder, which highlight its intensity, whereas the preferred orientation along the fiber axis of the films hides those reflections [29]. For BC, the triclinic unit cell Iα is the crystal form which prevails, and its three main peaks (100), (010), and (110) [21,40] are centered around 14.5, 16.8, and 22.7° 2θ, respectively. There are also two other peaks centered around 20.36 and 34.6° 2θ, and their Miller indices are ($11\bar{2}$) and ($11\bar{4}$), respectively. These peaks are also more clearly distinguished in the powder than in the film, and this can be due to the same phenomena observed in the BR and CS samples and mentioned by French [29].

The milling in the P_BR_BM_10 m sample produced a broadening in the amorphous contribution to the XRD data with respect to the film (Figure 5b), and there is also a decrease in intensity for peaks ($1\bar{1}0$), (110), and (200), with the peak (110) being the one that suffers the greatest compared to the film diffractogram. For peak (102), the increase in its intensity may be due to the loss of the preferred orientation of the sample caused by milling, as has been observed for other nanoscale materials [41]. Figure 6 shows that in the CS experiments, there are similar transformations to those of BR after 10 min of milling; however, the peak (110) in CS does not experience the greatest reduction compared to the diffractogram of the film (Figure 6b), where the relative decrease after milling is homogeneous for all peaks. It is also important to note that peaks ($1\bar{1}0$) and (110) appear to be closer in BR than in CS.

In BC, the Iα phase predominates [35]. For BC, the peaks (100) and (010), which are the counterparts to the peaks ($1\bar{1}0$) and (110), respectively, in cellulose Iβ, are more clearly resolved than in BR and CS. These peaks, like the lignocellulosic samples, decrease in intensity after 10 min milling (Figure 6b), for which the effect is more pronounced for peak (100). Additionally, milling for 10 min causes broadening of the amorphous peak, which indicates dismantling of the crystalline structure [31].

The accuracy of fitting of peak (004) (in cellulose Iβ) or ($11\bar{4}$) (in cellulose Iα) is not the same accuracy as for previous bands. In the 10 min milled powder of each sample, it can be seen that the fitted data in the range 30–35° 2θ do not have an optimal adjustment to the original data. This can be explained by the existence of small peaks in this region, which can be observed in the XRD data for a highly crystalline cellulose sample [29] and are not considered here, as adding more crystalline peaks would increase the complexity and uncertainty of the deconvolution method. For this reason, we suggested using five crystalline peaks [12].

Table 4 presents the CI values and the d-spacing for peak (200) (in Iβ cellulose) or (110) (in Iα cellulose) obtained for each sample. The films are the samples with the most crystalline percentages due to the orientation phenomena. Film BR has the lowest values for the CI (%) and d-spacing of plane (200), whereas Film BC is the sample with the highest CI (%) and the highest d-spacing for peak (110). For BR and the remaining samples, 10 min milling led to a decrease in CI (%) of about 50%, with BC being the most affected by the treatment, as shown in Table 4. Despite this behavior, the d-spacing of plane (200) for BR and CS and of plane (110) for BC does not appear to have changed, and the small differences between the crystal-size values of the films and the powders seems to be due to statistical error.

Table 5 shows reference values for CI (%) reported in the literature. The values calculated here are in the range of the measurements observed in previous works. The values reported for commercial cellulose in the table are the reference values for Avicel PH-101 cellulose. The values calculated using the Segal method for BR Film and BC Film are higher than those calculated here, which is believed to have occurred due to the method implemented, since the Segal method gives higher CI values than peak deconvolution methods [9].

The correction of the polarization factor and background subtraction helped to improve the accuracy of the results in the deconvolution [12]. The Fourier-series fitting also allowed for accurate consideration of the amorphous contribution, as can be seen in Figure 4. As shown in this work, the use of Fourier series means progress in the determination of the CI for cellulose from different sources, and it should be used when studying the inner structure of this material through DRX.

## 4. Conclusions

Amorphous cellulose was obtained at room temperature through the easily accessible method of sustained ball milling. When the crystalline structure was dismantled, the FTIR and XRD results, regardless of the type of cellulose, converged to a common profile.

The peak deconvolution method was used to calculate the cellulose crystallinity index; however, common peak functions do not adequately fit the amorphous profile, which affects the accuracy in determining the amorphous profile portion of the whole diffractogram pattern of a cellulose sample. In this study, we used a Fourier series to fit the amorphous cellulose XRD profile to obtain a higher R^2^. Fourier fitting in deconvolution allows acquisition of a model that more accurately represents the contribution of the amorphous profile and, thus, a more accurate CI calculation, with the values obtained being in the range reported in the literature. The effect of directionality in cellulose films was observed, leading to the development of improved protocols for sample preparation for analysis using XRD. The results obtained in this work can form the basis of a common groundwork to better study cellulose crystallinity and the relationship between the properties and inner structure of this biopolymer. This approach facilitates the possibility of more in-depth investigations for which more expensive or difficult-to-access techniques are not required.

## Figures and Tables

**Figure 1 polymers-14-05199-f001:**
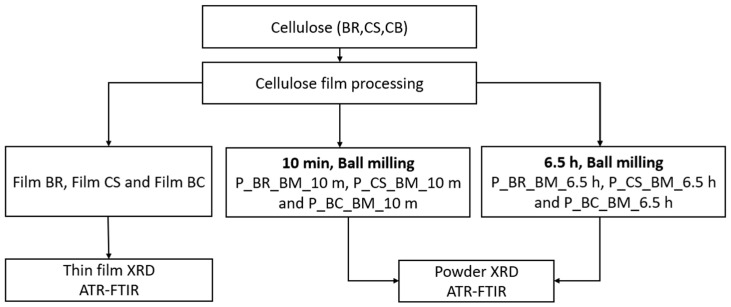
Flow process diagram. Cellulose from banana rachis (BR), a commercial sample (CS), and bacterial cellulose (BC) were used. Three samples were obtained from each cellulose type: an unmilled sample (film), a sample ball-milled for 10 min (P_BR_BM_10 m, P_CS_BM_10 m, and P_BC_BM_10 m), and one ball-milled for 6.5 h (P_BR_BM_6.5 h, P_CS_BM_6.5 h, and P_BC_BM_6.5 h).

**Figure 2 polymers-14-05199-f002:**
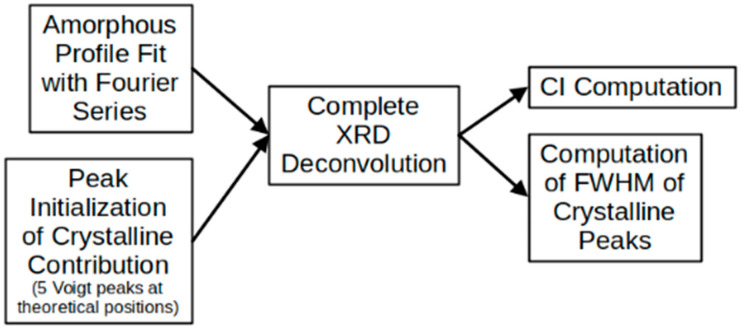
Diagram of main steps followed in the MATLAB routine to perform deconvolution of complete XRD profile and to obtain CI and FWHM (width) of crystalline peaks.

**Figure 3 polymers-14-05199-f003:**
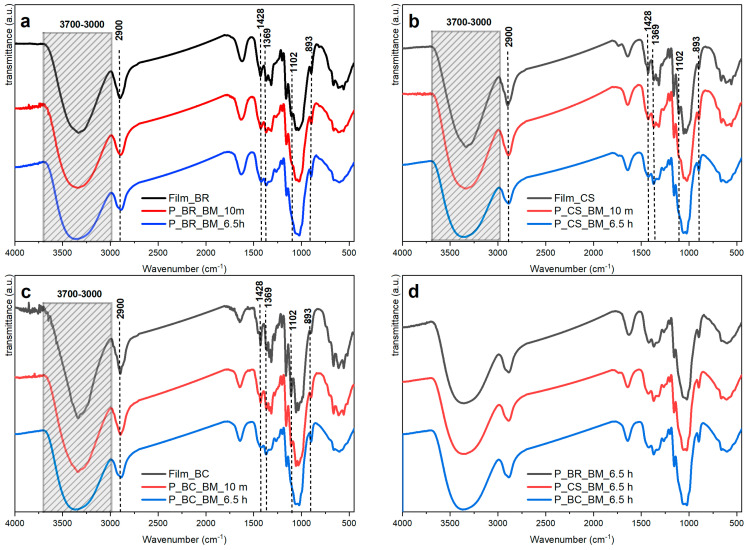
Infrared spectra of cellulose samples: (**a**) banana rachis (BR) milled at 10 min (P_BR_BM_10 m) and 6.5 h (P_BR_BM_6.5 h), and unmilled (Film BR); (**b**) commercial sample (CS) milled at 10 min (P_CS_BM_10 m) and 6.5 h (P_CS_BM_6.5 h), and unmilled (Film CS); (**c**) bacterial cellulose (BC) milled at 10 min (P_BC_BM_10 m) and 6.5 h (P_BC_BM_6.5 h), and unmilled (Film BC); and (**d**) BR, CS, and BC samples at 6.5 h milling time.

**Figure 4 polymers-14-05199-f004:**
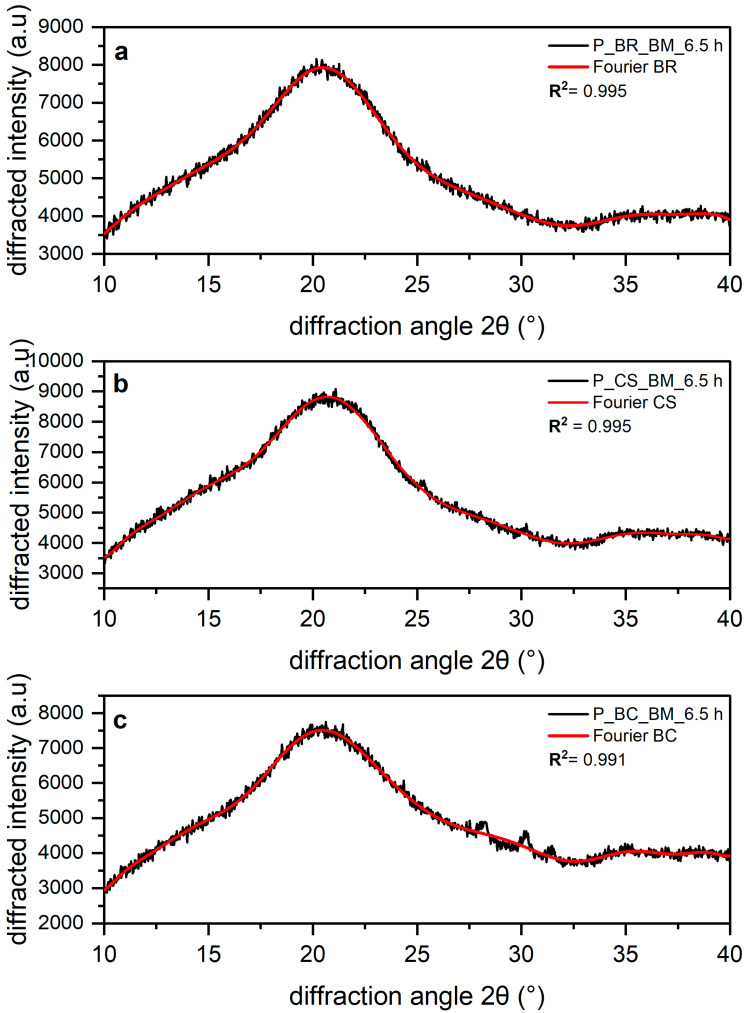
(**a**) Amorphous banana rachis (BR), (**b**) commercial sample (CS), and (**c**) bacterial cellulose (BC) XRD. The black line corresponds to the experimental result and the red line to data fit using the Fourier function.

**Figure 5 polymers-14-05199-f005:**
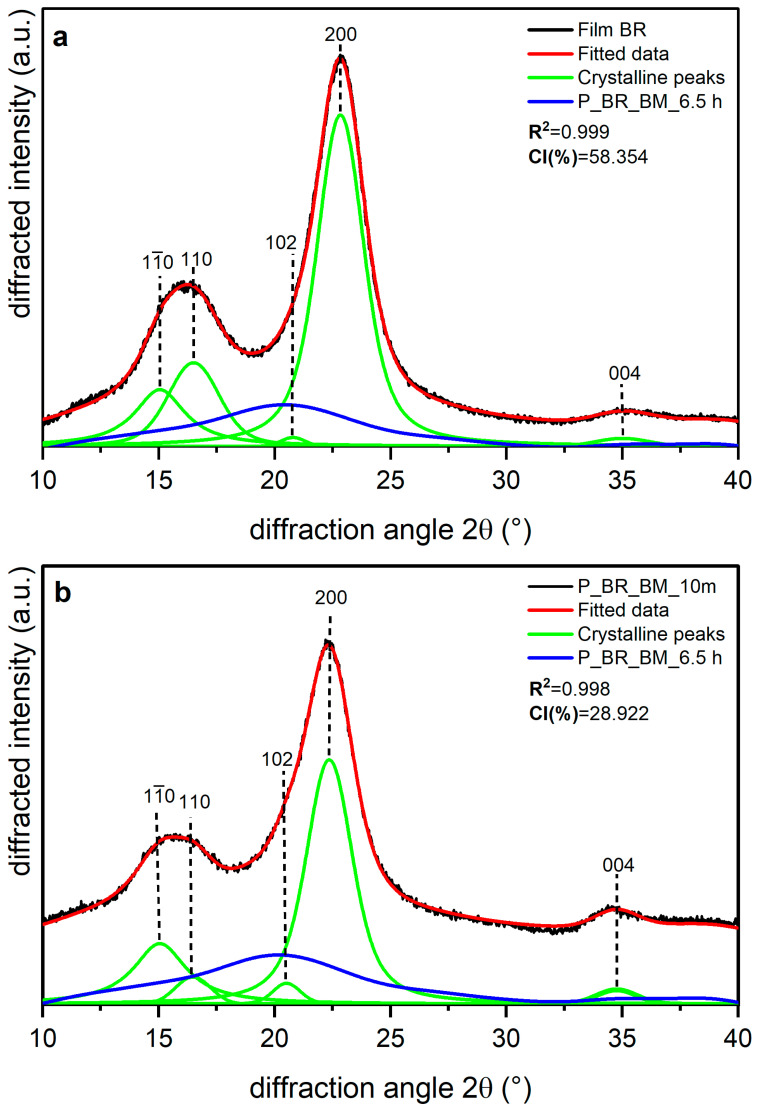
X-ray diffraction spectra recorded from banana rachis cellulose samples (BR): (**a**) unmilled (film BR), and (**b**) milled for 10 min (P_BR_BM_10 m) and 6.5 h (P_BR_BM_6.5 h). The indexation is that defined in [40].

**Figure 6 polymers-14-05199-f006:**
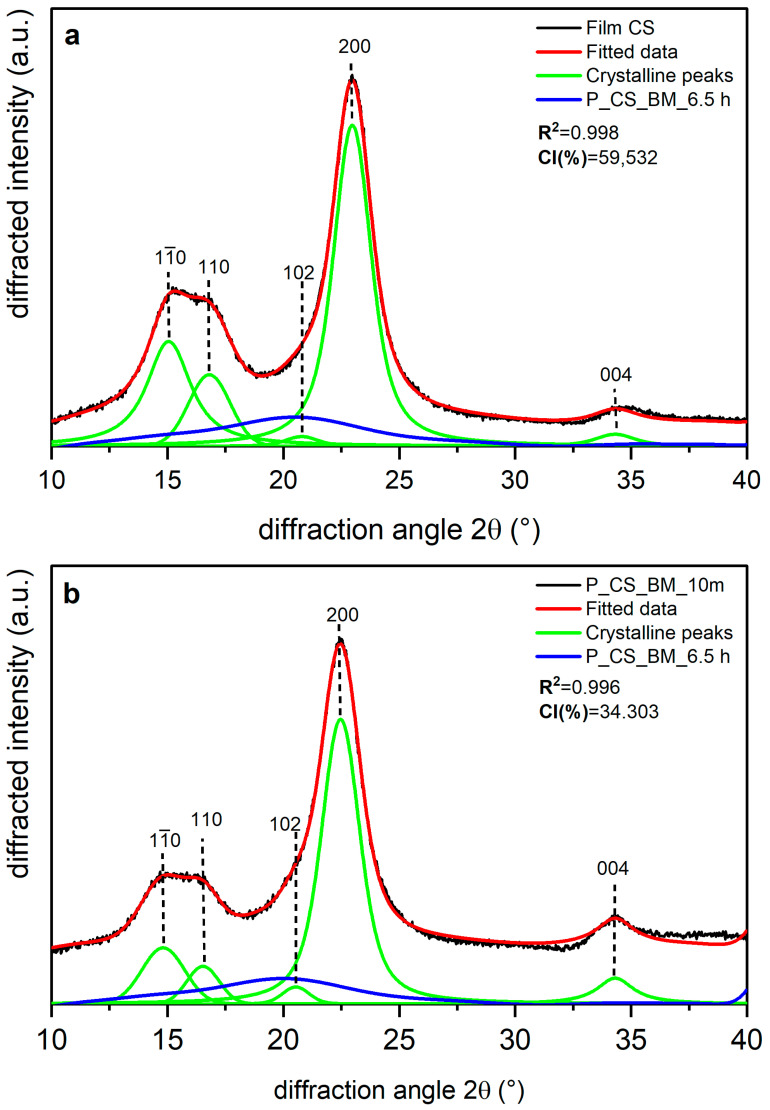
X-ray diffraction spectra recorded from commercial sample cellulose (CS): (**a**) unmilled (film CS), and (**b**) milled for 10 min (P_CS_BM_10 m) and 6.5 h (P_CS_BM_6.5 h). The indexation is that defined in [40].

**Figure 7 polymers-14-05199-f007:**
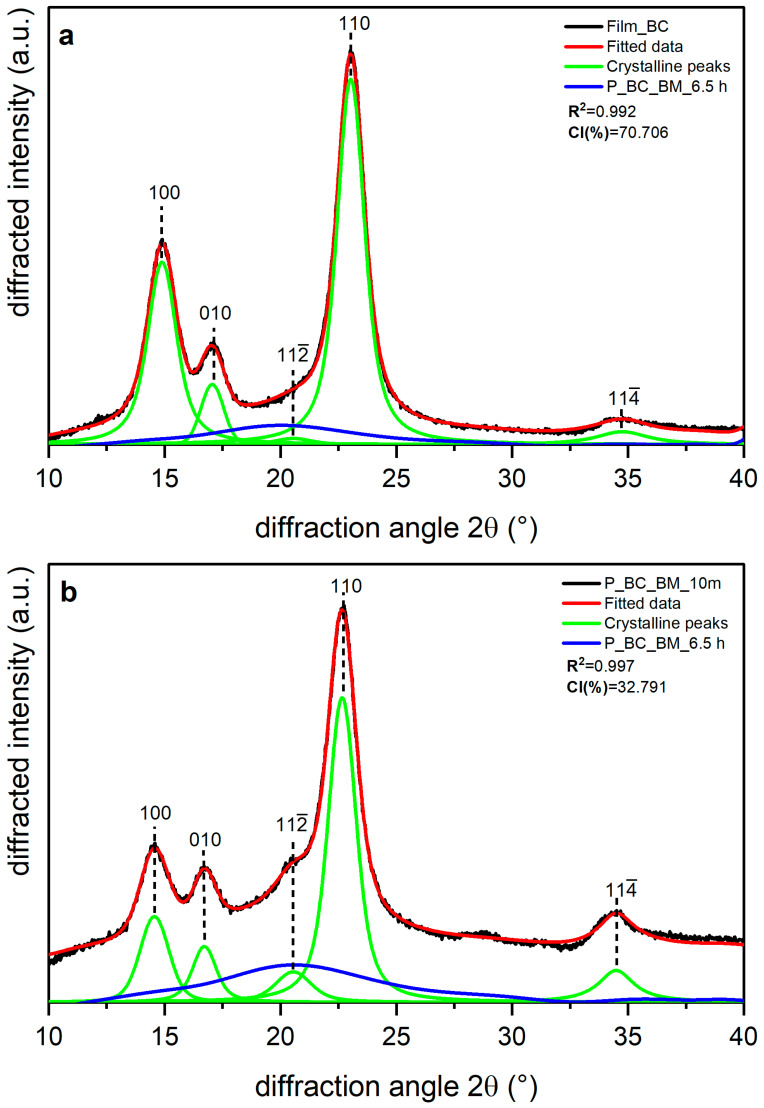
X-ray diffraction spectra recorded from bacterial cellulose (BC): (**a**) unmilled (film BC), and (**b**) milled for 10 min (P_BC_BM_10 m) and 6.5 h (P_BC_BM_6.5 h). The indexation is that defined in [40].

**Table 1 polymers-14-05199-t001:** Ball-milling treatment conditions.

Vibrational Frequency	30 Hz
Sample amount	1.5 g
Max. feed size (mm)	6 mm
Temperature	24 °C
Pre-cooling	None
Grinding cooling	None

**Table 2 polymers-14-05199-t002:** Main absorbances associated with changes in the crystalline structure of the BR, CS, and BC samples.

Absorbance (cm^−1^)	Discussion	References
3000–3700	Assigned to the symmetric and antisymmetric OH stretching of inter- and intramolecular hydrogen bonds. A broadening or shifting of the absorbances in the spectra presented in Figure 3a–c is related to scission of the inter- and intramolecular hydrogen bonds with progressively increased milling. This disruption is believed to be the cause of the apparent intensity reduction seen when the milling time is increased.	[35,36,37]
2900	Assigned to the extension of the CH and CH_2_ bond. It shows a significant reduction in intensity with increasing milling time for all samples.	[4]
1428	Assigned to CH_2_ scissoring motion or bending. It also shows a reduction in intensity with the increase in milling time for all samples.	[4,37]
1369	Assigned to CH bending. The absorbance of this band is unaffected by water adsorbed onto cellulose. A slight decrease in intensity is observed for this absorbance with increased milling time. It has been used for estimation of CI. This band and those at 1335 and 1315 cm^−1^ show the greatest progressive changes with the extension of ball milling.	[4]
1102	Association band analogous to those found near 1111 cm^−1^ in primary and secondary alcohols. Attributed to the hydrogen bonding effect on the vibrations of the skeleton surrounding the extension of the C–O bond. Reductions in absorbance intensity are associated with amorphous cellulose. It is important to emphasize that the IR spectrum of cellulose II is quite similar to that of the amorphous cellulose regarding this band.	[4]
893	Absorbance characteristic of C_1_–H bond stretching in β-bonds of glucose and the four atoms attached to it. A slight increase is observed with extended milling time, which would be expected if the oxygen atoms attached to C_1_ are involved in this vibration, and changes around the glycosidic linkage and in the hydrogen bonds may also affect its intensity.	[4,37]

**Table 3 polymers-14-05199-t003:** Fourier-series coefficients found in the fitting conducted in MATLAB.

Coefficient	Banana Rachis (BR)	Commercial Sample (CS)	Bacterial Cellulose (BC)
w	0.1293	0.1071	0.1049
a0	1.923 × 10^4^	1.763 × 10^6^	3.142 × 10^6^
a1	2.638 × 10^4^	2.944 × 10^6^	5.013 × 10^6^
b1	4629	−1.386 × 10^6^	−2.93 × 10^6^
a2	2.331 × 10^4^	1.641 × 10^6^	2.251 × 10^6^
b2	5994	−1.986 × 10^6^	−3.999 × 10^6^
a3	1.666 × 10^4^	4.328 × 10^5^	−4.97 × 10^4^
b3	8217	−1.672 × 10^6^	−3.068 × 10^6^
a4	1.037 × 10^4^	−1.754 × 10^5^	−8.87 × 10^5^
b4	6608	−9.482 × 10^5^	−1.46 × 10^6^
a5	5550	−2.509 × 10^5^	−6.769 × 10^5^
b5	4824	−3.562 × 10^5^	−3.653 × 10^5^
a6	2132	−1.301 × 10^5^	−2.659 × 10^5^
b6	2544	−7.663 × 10^4^	9603
a7	786.1	−3.613 × 10^4^	−5.408 × 10^4^
b7	1032	−4452	3.443 × 10^4^
a8	101.8	−4359	−3748
b8	270	993	7154
R^2^	0.9948	0.995	0.9915
RMSE	96.24	110.1	113.7

**Table 4 polymers-14-05199-t004:** CI (%) calculated for the different samples with background subtraction and intensity correction.

Sample	CI (%)	d-Spacing ((200)-Iα. (110)-Iβ) (nm)	R^2^
Film BR	58.354	3.578 3.594	0.999
P_BR_BM_10 m	28.922		0.998
Film CS	59.532	4.425 4.341	0.998
P_CS_BM_10 m	34.303		0.996
Film BC	70.706	6.069 5.974	0.992
P_BC_BM_10 m	32.791		0.997

**Table 5 polymers-14-05199-t005:** Reference values of cellulose CI.

Sample	CI Value (%)	Method	Sample Form	Reference
Banana Rachis	63	Segal	Film	[42]
Commercial sample	60	Deconvolution (Gauss peaks)	Avicel	[2]
Commercial sample	54	Deconvolution (Gauss peaks)	Avicel	[39]
Bacterial Cellulose	71	Deconvolution (Gauss peaks)	Film	[43]
	83	Segal	Film	[44]
	33.43 ± 3.16	Deconvolution (Voigt peaks and Fourier series)	Powder (10 min milled)	[12]
	86	Rietveld method	Film	[45]

## Data Availability

The data presented in this study are available on request from the corresponding author.

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
