# Peer review of "Use of Fourier Series in X-ray Diffraction (XRD) Analysis and Fourier-Transform Infrared Spectroscopy (FTIR) for Estimation of Crystallinity in Cellulose from Different Sources"

_polymers, 2022, doi:10.3390/polym14235199_

Round 1

Reviewer 1 Report

In general, the document is fine. Authors must improve figures 4, 5, 6, and 7.

Author Response

In general, the document is fine. Authors must improve figures 4, 5, 6, and 7.

R/ Figure 4, 5 ,6 and 7 were improved, according to the comments of the reviewer

Reviewer 2 Report

Experimental article "Use of Fourier series in X-ray diffraction (XRD) analysis and Fourier transform infrared spectroscopy (FTIR) for estimation of crystallinity in cellulose from different sources", dedicated to the description of the method for determining the crystallinity of cellulose from various sources, according to its main idea, proposed methods, discussion of the results obtained is absolutely consistent with the profile of the Polymers publication and is relevant in the field of structural chemistry of polymers, in this case, cellulose. The strength of this article is the citation of the publications of the modern founder of the structural chemistry of cellulose A.D. French, which in turn means that the authors adhere to fundamental ideas about the structure of cellulose. Interestingly, the authors set themselves the difficult task of comparing the capabilities of the proposed method on standard commercial wood pulp (formed in decades), experimental banana cellulose (BR) (formed within one year), and bacterial cellulose (formed in 8 to 16 days). The idea of complete amorphization of crystalline cellulose was used by the authors to substantiate the efficiency of the proposed method. But according to the reviewer, the authors ignored some works that prove that the method they propose cannot be called new. This does not mean that the article can be rejected, but it would be helpful for both authors and readers to have these views reflected in the article “Use of Fourier series in X-ray diffraction (XRD) analysis and Fourier transform infrared spectroscopy (FTIR ) for estimation of crystallinity in cellulose from different sources”.

Detailed notes are provided below:

1. The technique used in this work to deconvolve XRD data to estimate CI and full width at half maximum (FWHM) is not new in principle; it has been described in detail in a number of works, in particular, in the work of Sunkyu Park et al.

Sunkyu Park, John O Baker, Michael E Himmel1, Philip A Parilla and David K Johnson R Cellulose crystallinity index: measurement techniques and their impact on interpreting cellulase performance Biotechnology for Biofuels 2010, 3:10, http://www.biotechnologyforbiofuels.com/content/3/1/10

The following statement pertains to the same question:

The choice of the profile function, as a rule, is determined by the geometry of registration of the diffraction pattern, and in the software used by various researchers, in particular, implementing the Rietveld method, in addition to the three functions indicated by the authors, there are at least three more accurately describing the profile of reflections from crystalline cellulose.

2. The idea of ​​using X-ray diffraction patterns of amorphous cellulose is also not new, so the authors need to link their results with the review by M. Ioelovich. Compare your results (in particular, the drawings of amorphous celluloses) with those in M. Ioelovich's review.

Ioelovich M. Preparation, Characterization and Application of Amorphized Cellulose-A Review. Polymers 2021, 13, 4313. https://doi.org/10.3390/polym13244313

3. The deconvolution technique is actively used for cellulose from a wide range of cellulose sources, here are some modern examples:

Torlopov MA, Mikhaylov VI, Udoratina EV et al (2018) Cellulose nanocrystals with different length-to-diameter ratios extracted from various plants using novel system acetic acid/phosphotungstic acid/octanol-1. Cellulose. 25:1031–1046. https://doi.org/10.1007/s10570-017-1624-z  

Bogolitsyn, K., Parshina, A., & Aleshina, L. (2020). Structural features of brown algae cellulose. Cellulose, 27(17), 9787-9800. https://doi.org/10.1007/s10570-020-03485-z

Aleshina L.A., Gladysheva E.K., Budaeva V.V., et al. Crystallogr. Rep. 63 (6), 955 (2018). https://doi.org/10.1134/S1063774518050024

4. It is known that X-ray diffraction (XRD) analysis more deeply reveals the structural features of cellulose from plant sources (the presence of only I betta allomorph) and bacterial cellulose (the presence of two allomorphs I betta and I alpha with the advantage of I alpha), but the authors completely are not related to this topic.

Author Response

  1. The technique used in this work to deconvolve XRD data to estimate CI and full width at half maximum (FWHM) is not new in principle; it has been described in detail in a number of works, in particular, in the work of Sunkyu Park et al.

R/ We agree with the reviewer, we are acknowledge that the deconvolution technique used in this work to estimate CI and FWHM is not new, we have cited the work of Sunkyu Park et al. in the sixth line of the introduction. The novelty of our work is the Fourier fitting of the amorphous profile, it allowed to more effectively model the contribution of the amorphous profile in the XRD deconvolution. This method, to the best of our knowledge, has only been used by Yao et al. in 2020, and has contributed to produce a more satisfactory fitting of the amorphous profile than common peak functions such as Lorentz and Gaussian, which has been a problem when estimating crystallinity of cellulose samples. It is also important to note, that Yao et al. used a dedicated commercial software while we developed an algorithmin MATLAB to perform the deconvolution, which allows a higher flexibility in the results.

Probably the misunderstanding was caused by in the line 40 in the abstract. This line and line 96 in the introduction, were changed to avoid this kind of inconvenience. 

Line 96 was changed: Fourier series were used to fit the amorphous profile, which has been found to be effective in modelling the amorphous contribution for different kinds of cellulose and reference of Yao [12] was added.

2.  The idea of using X-ray diffraction patterns of amorphous cellulose is also not new, so the authors need to link their results with the review by M. Ioelovich. Compare your results (in particular, the drawings of amorphous celluloses) with those in M. Ioelovich's review.

Ioelovich M. Preparation, Characterization and Application of Amorphized Cellulose-A Review. Polymers 2021, 13, 4313. https://doi.org/10.3390/polym13244313

R/ Different authors have used diffraction patterns of amorphous cellulose, but the breakthrough in our approach is that we are using a differentiated approach to the amorphous profile changing the cellulosic source to match the material under study, before proceeding to develop the deconvolution. Not the same amorphous profile for cellulose from different sources.

We have read the recommended reference, and it was included in the reviewed version, line 119. This reference supports the process in which the grinding amorphize the crystalline structure of cellulose, as it is the most widespread physical method for this purpose.

Ielovich reference [21] was added in line 119

3. The deconvolution technique is actively used for cellulose from a wide range of cellulose sources, here are some modern examples:

Torlopov MA, Mikhaylov VI, Udoratina EV et al (2018) Cellulose nanocrystals with different length-to-diameter ratios extracted from various plants using novel system acetic acid/phosphotungstic acid/octanol-1. Cellulose. 25:1031–1046. https://doi.org/10.1007/s10570-017-1624-z

Bogolitsyn, K., Parshina, A., & Aleshina, L. (2020). Structural features of brown algae cellulose. Cellulose, 27(17), 9787-9800. https://doi.org/10.1007/s10570-020-03485-z

Aleshina L.A., Gladysheva E.K., Budaeva V.V., et al. Crystallogr. Rep. 63 (6), 955 (2018). https://doi.org/10.1134/S1063774518050024

R/ Thanks to the reviewer for the relevant references about the deconvolution technique. All of them were used to improve the reviewed version and to compare the reported results 

Torpolov reference [13] was added in lines 67 and 230.

Bogolitysin reference [32] was added in line 230.

Aleshina reference [46] was added in Table 5.

4. It is known that X-ray diffraction (XRD) analysis more deeply reveals the structural features of cellulose from plant sources (the presence of only I betta allomorph) and bacterial cellulose (the presence of two allomorphs I betta and I alpha with the advantage of I alpha), but the authors completely are not related to this topic.

R/ Considering this comment, as described above, the reviewed version includes additional information about the predominant crystal forms of cellulose isolated from plants, and bacterial cellulose:

Line 306 was modified to add “In cellulose of land plants, such as BR and CS, the predominant crystal form is monoclinic Iβ.”

Line 321 was modified: “For BC, the triclinic unit cell Iα is the crystal form which prevails, as evidenced by its three main peaks.”

Ielovich [21] reference was added in line 307 y 322

Round 2

Reviewer 2 Report

Dear authors, thank you for understanding my comments and I apologize that I missed the work of Park (reference 9) in the original version of the article. This is my comment 1.

Your work, as I emphasized earlier, is very relevant and attracts attention with the complexity of the goal: to describe the XRD analysis of cellulose from various plant sources, plus to compare the results of XRD analysis of plant and bacterial cellulose.